# Can the Stream Quantification Tool (SQT) Protocol Predict the Biotic Condition of Streams in the Southeast Piedmont (USA)?

**Sara Donatich [1],\*** , **Barbara Doll [1,2]** , **Jonathan Page [1]** and **Natalie Nelson [1]**

1   Department of Biological and Agricultural Engineering, North Carolina State University, Raleigh, NC 27606, USA; bdoll@ncsu.edu (B.D.); jlpage3@ncsu.edu (J.P.); nnelson4@ncsu.edu (N.N.)
2   North Carolina Sea Grant, Raleigh, NC 27606, USA
\*   Correspondence: srdonati@ncsu.edu

**Abstract:** In some states, the Stream Quantification Tool (SQT) has been adopted to quantify functional change of stream mitigation efforts. However, the ability of the SQT protocol to predict biological function and uphold the premise of the Stream Functions Pyramid (Pyramid) remains untested. Macroinvertebrate community metrics in 34 headwater streams in Piedmont, North Carolina (NC, USA) were related to NC SQT protocol (version 3.0) factors and other variables relevant to ecological function. Three statistical models, including stepwise, lasso, and ridge regression were used to predict the NC Biotic Index (NCBI) and Ephemeroptera, Plecoptera, and Trichoptera (EPT) richness using two datasets: 21 SQT variables and the SQT variables plus 13 additional watershed, hydraulic, geomorphic, and physicochemical variables. Cross-validation revealed that stepwise and ridge outperformed lasso, and that the SQT variables can reasonably predict biology metrics ($R^2$ 0.53–0.64). Additional variables improved prediction ($R^2$ 0.70–0.88), suggesting that the SQT protocol is lacking metrics important to macroinvertebrates. Results moderately support the Pyramid: highly predictive ridge models included metrics from all levels, while highly predictive stepwise models included metrics from higher levels, and not watershed hydrology. Reach-scale metrics were more important than watershed hydrology, providing encouragement for projects limited by watershed condition.

**Keywords:** streams/rivers; restoration; ecological/stream function; benthic macroinvertebrates; stream quantification tool (SQT); stream functions pyramid framework; compensatory mitigation

## 1. Introduction

The emphasis on restoring ecological functions is an evolving field in stream and river restoration research [1–4] and practice. Functions characterize an ecosystem's processes, roles, services, or state of trajectory [5].

To improve the success of restoration, researchers and practitioners have sought to make connections between landscape and instream variables and biological condition [6]. Schueler [7] compiled several studies that associate declines in diversity of aquatic insects and fish, and instream habitat quality, when watershed imperviousness exceeds 10%–15% e.g., [8–12]. Riparian deforestation leads to loss of large woody debris, leaf litter, and organic dissolved inputs, elevates stream temperatures, and narrows stream channels, due to encroachment of herbaceous vegetation that would have otherwise been shaded out by the forest canopy [13,14]. A study of 16 streams in eastern North America concluded that forested channels exhibited lower average channel velocities, higher bed roughness, greater abundance of macroinvertebrates, and a greater amount of organic matter processing

compared to deforested channels [15]. Researchers have predicted macroinvertebrate assemblages with watershed-scale variables, such as catchment-wide geology and land cover [16,17] and reach-scale variables, such as riparian corridor land use change [17–19].

Research on methods to implement and evaluate the functional uplift resulting from restoration actions is a relatively new path of study [2] that involves identifying functions that support desired ecosystem services and how best to recover those lost functions [20]. Recent studies have evaluated the tools for gauging ecological change resulting from restoration efforts [21–23]. Other studies have focused on evaluating the response of macroinvertebrates and fish communities to restoration efforts [24–26]. In an assessment of 79 stream reaches across North Carolina (NC), Doll et al. [24] showed that EPT taxa (Ephemeroptera, Plecoptera, and Trichoptera) were positively correlated with accessible floodplain width (i.e., entrenchment ratio), substrate size, and mean bankfull depth, and negatively correlated with width-to-depth ratio and sinuosity. Results indicated that deliberate site selection, restoration activities, and design can optimize biotic condition and ecological function. Further, results indicated that larger streams with steeper valleys, larger substrate, and undeveloped watersheds were expected to have higher numbers of EPT taxa [24]. At 28 predominantly agricultural watershed sites in Ohio, D'Ambrosio et al. [25] also found that macroinvertebrate communities were driven by floodplain connection, as well as stream slope and size. In a study of 31 wadeable Piedmont Georgia streams, Walters et al. [17] found that macroinvertebrate metrics were best predicted by urban cover, specific conductivity, fines in riffles, and local relief.

D'Ambrosio et al. [26] found that fish assemblages in 32 Ohio streams were influenced by watershed- and reach-scale factors, including stream order, percent (%) wooded riparian zone, drainage area, instream cover quality, substrate quality, average substrate size, stream slope, stream size (i.e., cross-sectional area), and width of the flood prone area. These results indicated that site selection, geomorphic features, and instream habitat should be carefully considered for biological restoration projects. Similarly, Walters et al. [17] found fish metrics were best predicted by embeddedness, turbidity, stream slope, and forest cover. In a study of 21 Piedmont streams in Alabama, Helms et al. [27] found that measures of fish assemblage and crayfish size were strongly predicted by watershed size (i.e., drainage area) and geomorphic channel dimensions related to stream size, including bankfull cross-sectional area, width, mean depth, and discharge. Conversely, Walters et al. [17] found that both fish and macroinvertebrate metrics correlated poorly with measures of stream size, such as bankfull channel width, depth, and cross-sectional area, and drainage area and discharge.

Despite knowledge advancements in stream function relationships, measuring differences between pre- and post-restoration conditions and targeting restoration actions that improve biology is complicated by the interrelatedness of function variables [24], such that effects on one function can initiate a cascade of effects on other functions [28]. Understanding the relationships between these physical, chemical, and biological variables is challenged by the drastic disturbance often associated with restoration, such as changes in channel geometry, perturbation of soils, and inclusion of constructed structures [29].

The United States (US) requires compensatory mitigation projects to restore natural or historic aquatic resource functions [30]. The US defines functions as physical, chemical, and biological processes of ecosystems, which are also considered services when these processes benefit humans [30]. To help practitioners implement functional restoration projects, in 2006, Fischenich [28] proposed a functional framework that identified and organized a hierarchy of basic stream and riparian corridor functions described in the literature. Building on Fischenich's framework [28], in 2012, the Stream Functions Pyramid Framework (Pyramid Framework) was developed [31]. The Pyramid Framework asserts that functions are interrelated such that they build on each other in a specific order, whereby watershed hydrology is the fundamental support for hydraulic, geomorphic, physicochemical, and biological functions, and biology relies on all underlying functions (Figure 1). The Pyramid Framework was designed to help regulators and practitioners visualize how physical watershed and fluvial hydromorphological factors support physicochemical and biological functions and identify which

factors must be addressed to improve these higher-level functions. The Pyramid Framework also identifies specific parameters and metrics (indicators of function or direct measures of function) that can be measured for pre- and post-restoration condition monitoring.

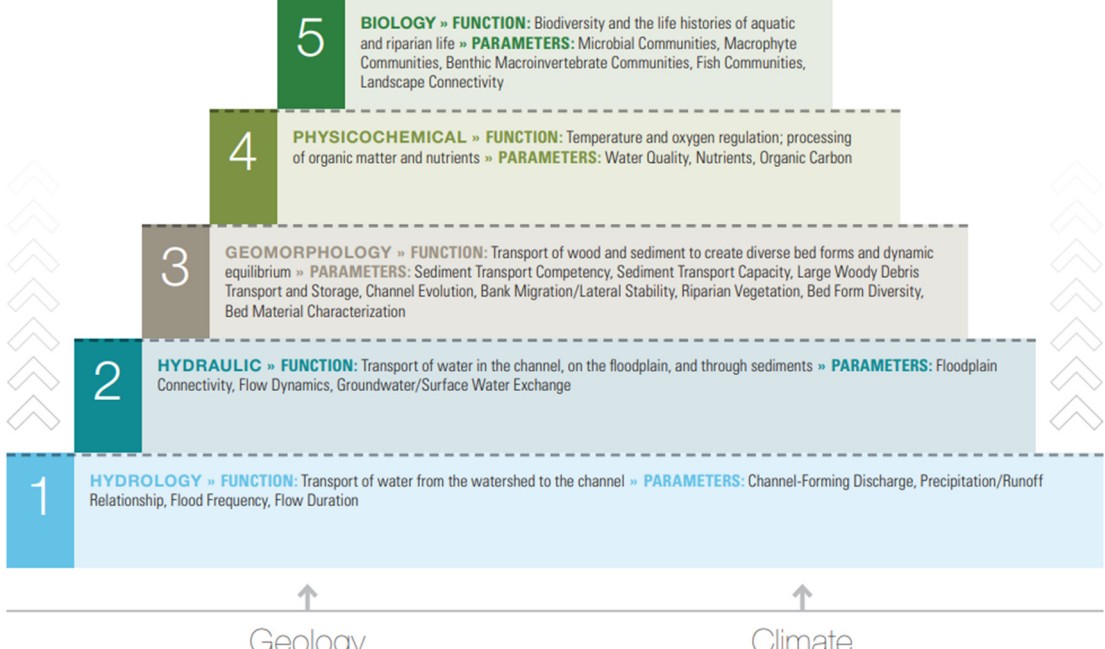

**Figure 1.** Stream Functions Pyramid Framework, depicting function category statements and function parameters for all five 5 levels of the Pyramid [31]. Reproduced with permissions from Harman et al., A Function-Based Framework for Stream Assessment and Restoration Projects, US Environmental Protection Agency, Office of Wetlands, Oceans, and Watersheds, Washington, DC, EPA 843-K-12-006, 2012.

The Pyramid Framework is the basis of the Stream Quantification Tool (SQT) [32], a stream mitigation crediting tool in use in Minnesota, Tennessee, Wyoming, and Georgia [33–36] and in development in Colorado, Michigan, and Alaska to date [37]. The first SQT was developed for NC [32] but is not formally adopted as a mitigation crediting tool at this time. The SQT aims to support mitigation site selection, determine a site's restoration potential, and quantify functional change associated with restoration activities to aid in mitigation debit and credit determination [32]. The SQT consists of a conditions assessment and an excel workbook tool that uses embedded reference (performance) curves to assign function index scores to measured field and desktop values. Function index scores are multiplied by linear feet of pre-restoration, proposed, and/or restored stream to calculate "functional feet," which are used to determine the quantity and quality of impacted, proposed, and/or restored streams and inform credit and debit awards. However, the relationships proposed in the Pyramid Framework and that underpin the SQT have not yet been statistically evaluated. Further, the ability of the SQT conditions assessment protocol (SQT protocol) to predict biological function is untested.

To address gaps in our understanding of the universal applicability of the Pyramid Framework as a schema for explaining biological function restoration, we apply the NC SQT protocol to test its predictive ability. We hypothesize that the Pyramid Framework is generalizable, in that hydrologic variables explain the most variance in biological function variables, while other higher-level variables (e.g., hydraulics, geomorphology, and physicochemistry) explain relatively less variance. The specific objectives of this study are to: (1) document stream functional data in the Piedmont ecoregion of NC (USA); (2) evaluate the SQT protocol's ability to predict biological function; and (3) determine variables

most important to predicting biological function. Results from objectives (2) and (3) will evaluate the hierarchical premise of the Pyramid Framework, which in theory underlies the SQT protocol. Identifying improvements to the SQT protocol for NC and other states is also an anticipated outcome of this effort.

## 2. Materials and Methods

### 2.1. Site Selection

The NC SQT (version 3.0) protocol [32] was applied to 34 headwater streams in the Piedmont ecoregion of NC, USA in 2017–2018 (n = 31) and 2019 (n = 3). Study sites were non-randomly selected from a database of reference reaches used by consulting firms for restoration design, a database of biological reference sites monitored by the NC Department of Environmental Quality (NC DEQ), available restoration project documents and monitoring data, professional recommendations, and physical access. Study sites included 18 geomorphic reference sites, 1 biological reference site, 9 restored sites, and 6 degraded sites (Figure 2; Table 1). A diversity of stream conditions was sampled to capture the variability in functional performance and to establish reasonable expectations for the potential of restored streams. All restored sites were at least five years old prior to assessment and perceived as moderately to highly successful on the basis of monitoring data and exhibited features similar to reference streams: stable streambanks, diverse bedforms (e.g., riffle, pool), substantial vegetative cover, and overbank flooding [22]. The degree of restoration varied from enhancement (grading of floodplain benches and addition of rock and/or log structures) to channel relocation and/or reconfiguration of channel dimensions. Watersheds ranged from 0.2 to 22.0 km$^2$. Watershed land use varied amongst reference, restored, and degraded sites from deciduous forest, herbaceous vegetation, agricultural fields, turfgrass, to urban–exurban development (Table 1). Stream order [38] ranged from one to three; one stream was characterized as intermittent.

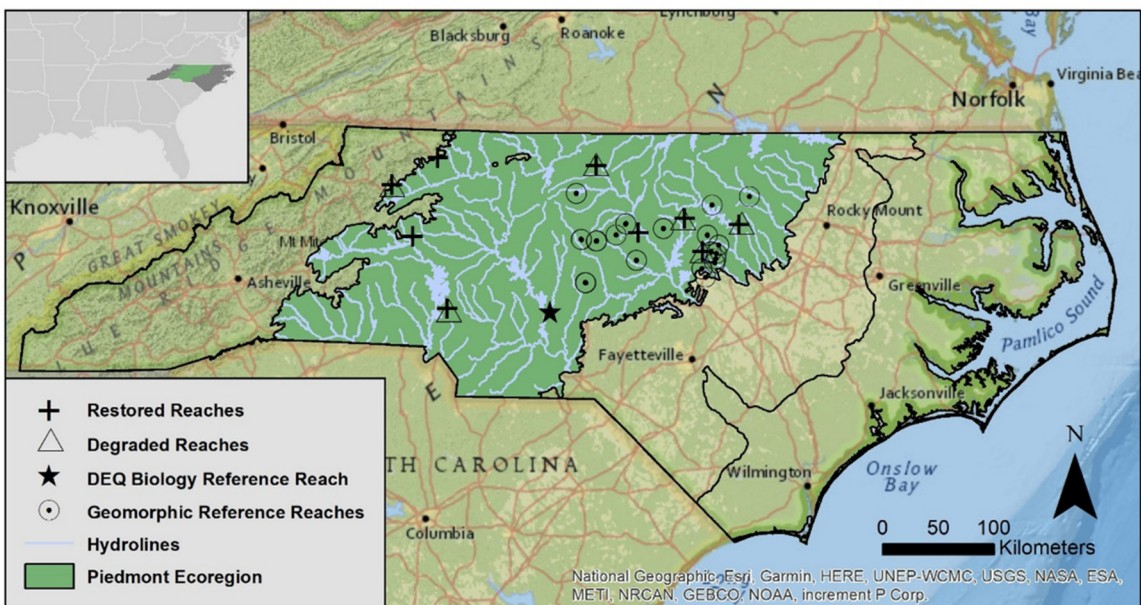

**Figure 2.** Location of 34 stream assessment sites in the Piedmont ecoregion, NC (Southeast, USA).

**Table 1.** Select attributes for restored (n = 9), degraded (n = 6), geomorphic reference (n = 18) study sites and the biological reference (n = 1) study site located in the Piedmont ecoregion, NC (Southeast, USA).

| Attribute | Restored | | Degraded | | Geomorphic Ref. | | Biological Ref. |
|---|---|---|---|---|---|---|---|
| | (Range; Arithmetic mean) | | | | | | (Value) |
| Drainage Area (km$^2$) | 1.0–22.0; | 6.2 | 0.5–9.8; | 3.4 | 0.2–21.3; | 4.3 | 8.9 |
| Impervious Cover (%) | 0.0–40.1; | 12.0 | 0.2–47.8; | 21.4 | 0.1–18.1; | 4.7 | 0.0 |
| Developed Cover (%) | 0.0–98.3; | 40.3 | 2.1–98.9; | 61.1 | 0.3–96.9; | 23.2 | 1.1 |
| Forested Cover (%) | 1.7–98.2; | 38.5 | 0.7–76.9; | 26.6 | 3.1–94.8; | 53.3 | 96.4 |
| Agricultural Cover (%) | 0.3–55.6; | 16.6 | 0.0–23.4; | 8.4 | 0.0–51.4; | 16.5 | 0.6 |
| Reach slope, $S_{avg}$ (%) | 0.2–4.6; | 1.0 | 0.3–2.1; | 0.9 | 0.4–2.3; | 0.9 | 0.9 |
| Median Substrate Diameter, $D_{50}$ (mm) | 0.6–19.3; | 5.0 | 0.6–19.3; | 6.2 | 0.4–77.0; | 26.9 | 52.5 |

## 2.2. Data Collection

Twenty-three (23) of the 26 hydrological, hydraulic, geomorphic, physicochemical, and biological variables included in the NC SQT protocol [32] were measured (SQT dataset). Aggradation ratio was not measured because it is recommended for sites where aggradation is evident [32], which did not apply to all stream sites. The NC Index of Biotic Integrity for fish was not assessed due to funding limitations. Bed material was characterized, but study limitations prohibited the data processing proposed by Harman and Jones [32], which requires comparative reference reaches for each site. Thus, substrate data were considered a non-SQT variable for study analyses. To determine which variables are most important to macroinvertebrate biota (Objective 3), 13 additional hydrological, hydraulic, and physicochemical variables known to affect macroinvertebrate biota were also measured (hereafter referred to as the SQT+ dataset; Table 2). Reaches were selected in the field via visual assessment by walking the stream to confirm similar stability conditions, streambank and riparian vegetation type, longitudinal slope, and bed material composition [32]. Each reach had a length that was at least 20 times bankfull channel width ($W_{bkf}$) for geomorphic relationships [39].

**Table 2.** Summary of data for 34 Piedmont stream sites. *Italicized* variables are additional variables, not included in the SQT protocol.

| Variable | Units | Arithmetic Mean ± SD | Range |
|---|---|---|---|
| Hydrology | | | |
| Runoff curve number (CN) | - | 69.1 ± 8.1 | 51–84 |
| Concentrated flow points | #/305 m | 1.1 ± 1.8 | 0–5.9 |
| Soil compaction | cm | 24.8 ± 12.5 | 6.8–53.4 |
| *Drainage area (DA)* | km$^2$ | 4.8 ± 5.5 | 0.2–22.0 |
| *Impervious cover* | % | 9.4 ± 12.5 | 0.0–47.8 |
| *Developed cover* | % | 33.8 ± 36.4 | 0.0–98.9 |
| *Forested cover* | % | 45.9 ± 30.8 | 0.7–98.2 |
| *Agricultural cover* | % | 14.6 ± 17.1 | 0.0–55.6 |
| Hydraulics | | | |
| Bank height ratio (BHR) | - | 1.4 ± 0.5 | 1.0–2.8 |
| Entrenchment ratio (ER) | - | 6.2 ± 4.4 | 1.2–19.3 |
| *Bankfull channel width ($W_{bkf}$)* | m | 5.4 ± 2.6 | 2.2–13.8 |
| *Mean bankfull depth ($d_{bkf}$)* | m | 0.4 ± 0.1 | 0.2–0.9 |
| *Bankfull area ($A_{bkf}$)* | m$^2$ | 2.5 ± 1.8 | 0.6–9.1 |
| *Average channel slope ($S_{avg}$)* | % | 1.0 ± 0.8 | 0.2–4.6 |
| *Width-to-depth ratio (W/D)* | - | 12.8 ± 4.7 | 6.9–22.6 |

**Table 2.** *Cont.*

| Variable | Units | Arithmetic Mean ± SD | Range |
|---|---|---|---|
| Geomorphology | | | |
| Riffle & run extent (% Riffle) | % | 59.3 ± 14.9 | 33.9–95.5 |
| Pool spacing ratio | - | 6.3 ± 4.2 | 0–23.08 |
| Pool depth ratio | - | 2.8 ± 0.9 | 1.4–5.4 |
| Sinuosity | - | 1.3 ± 0.2 | 1.0–1.7 |
| Riparian buffer width | m | 113.7 ± 131.2 | 0–619 |
| Streambank erosion | % | 16.5 ± 14.3 | 0–55 |
| Bank erosion hazard index (BEHI) | Index | 29.6 ± 6.7 | 20–45 |
| Near-bank stress (NBS) | Index | 2.9 ± 1.2 | 1–5 |
| Basal area | $m^2$/ha | 20.5 ± 13.5 | 0–58 |
| Large woody debris index (LWDI) | Index | 184.6 ± 147.1 | 10–542 |
| Canopy cover | % | 84.4 ± 22.9 | 2–97 |
| $D_{50}$ | mm | 18.2 ± 21.8 | 0.4–77.0 |
| $D_{84}$ | mm | 75.3 ± 106.9 | 3.0–618.0 |
| Physicochemistry | | | |
| Total nitrogen (TN) | mg/L | 1.1 ± 1.0 | 0.18–5.85 |
| Total phosphorus (TP) | mg/L | 0.1 ± 0.0 | 0.01–0.20 |
| Summer temperature | deg. C | 22.1 ± 2.1 | 17.1–25.7 |
| Fecal coliform | cfu/100 mL | 964.7 ± 1498.4 | 10–5909 |
| Shredder taxa (% Shredders) | % | 9.8 ± 11.7 | 0.0–50.0 |
| *Specific conductivity* * | uS/cm | 122.3 ± 72.2 | 34.1–370.6 |
| Biology | | | |
| EPT richness | # & Index | 11.1 ± 9.5 | 0–32 |
| NC biotic index (NCBI) | Index | 5.3 ± 1.4 | 2.03–8.43 |
| *Total taxa richness* | # | 29.1 ± 13.9 | 6–54 |

* Specific conductivity is included in the SQT's Best Management Practice (BMP) Routine, which was not tested in this study. Therefore, it is considered an additional metric.

### 2.2.1. Hydrology

The United States Geological Survey's (USGS) StreamStats version 4.3.0 tool was used to delineate watersheds and determine land use cover (% impervious, developed, agricultural, and forested cover) on the basis of the 2011 National Land Cover Database (NLCD). Soil survey data was retrieved for the target watersheds from Natural Resources Conservation Service [40] and watersheds were divided by United States Department of Agriculture (USDA) hydrologic soil groups (A, B, C, or D) and USGS NLCD grid codes [41]. Using combined soils and land use data, runoff curve numbers (CNs) were assigned following Cronshey et al. [42]. Area-weighted, composite runoff CNs were calculated for each watershed using ArcGIS Desktop 10.3.1 software [43]. The SQT protocol requires the calculation of area-weighted, composite runoff CNs for upstream and lateral watersheds, with respect to the reach. For this study, composite CNs for the entire watershed were used for analyses. Concentrated flow points (e.g., ephemeral erosional gullies, storm drains, outfalls/pipes, and agricultural ditches) were counted along the stream reach and normalized over a 305-m reach length (1000 ft) [32]. Soil compaction, or depth to compacted layer, was measured within four plots along the right and left bank floodplain of the reach using a Dickey-John 76.2-cm Steel Soil Compaction Penetrometer, which measures up to 3447 Kpa (500 PSI) [32].

### 2.2.2. Hydraulics

A robotic total station was used to collect all cross-section and longitudinal data along the reach ("Detailed Method" proposed by Harman and Jones [32]). For each site, at least three representative riffle cross-sections were surveyed to establish a range of dimensions. For cross-sections, elevations were collected at inflection points across the floodplain and within the channel. All survey data was processed in AutoCAD Civil3D [44] and RIVERMorph 5.2.0 [45], and used to calculate average channel slope ($S_{avg}$), bankfull width ($W_{bkf}$), mean bankfull depth ($d_{bkf}$), bank height ratio (BHR), entrenchment ratio (ER), and width-to-depth ratio (W/D) [46]. BHR values weighted by riffle length were used in statistical analyses [32].

### 2.2.3. Geomorphology

A robotic total station was used to collect all longitudinal data along the reach ("Detailed Method" proposed by Harman and Jones [32]). Elevations for the thalweg, water surface, bankfull, and top of bank were collected at all key channel bedform features (i.e., head of all riffles, runs, pools, and glides, and the deepest part of the pool [max pool]). A representative pool spacing value was calculated by dividing the distance between sequential geomorphic and significant pools by bankfull width, using all pools or sometimes only representative pools along the reach [32]. Pool depth ratio was calculated by dividing max pool by mean riffle bankfull depth for every pool feature along the reach [32]. Sinuosity was calculated following Rosgen [46]. Percent (%) riffle was calculated by adding each riffle and run section, measured from head of riffle to head of pool, and dividing by the total reach length [32].

Bank Erosion Hazard Index (BEHI) and Near-Bank Stress (NBS) were measured at every meander bend and eroding bank along the reach. NBS was visually classified using the Rosgen Level I method [46]. The SQT protocol combines the dominant (mode) BEHI and NBS values—category scores describing the greatest percentage of stream bank—into one metric [32]. For statistical analyses, BEHI and NBS values were not combined. Percent (%) streambank erosion was measured for the reach by adding the total length of eroding banks and dividing by the total length of banks, and multiplying by 100 [32].

Quality and quantity of large woody debris (LWD) and log jams within the channel were assessed along 100 m (328 ft) of reach visually estimated to yield the highest LWD score, using the Large Woody Debris Index (LWDI) [32,47,48]. If more than 100 m of stream was assessed, results were normalized to 100 m. Most sites exhibited adjacent, mature deciduous forest, so basal area, in lieu of stem density, was measured for four 10-m plots along the left and right bank floodplain following US Army Corps of Engineers Wilmington District [32,49]. Floodplain canopy cover was measured at four cardinal locations within four plots on both sides of the floodplain along the reach with a Densiometer [50] during the growing season, following Harman and Jones [32]. Buffer width was measured to the edge of the riparian community or valley by placing 30.5-m (100-ft), perpendicular transects along the reach using the World Imagery Basemap [51] in ArcGIS Desktop 10.3.1 software [43] following Harman and Jones [32]. The median substrate diameter ($D_{50}$) and 84th percentile substrate diameter ($D_{84}$) were calculated from a modified Wolman Pebble Count [46].

### 2.2.4. Physicochemistry

Organic carbon content was estimated using the % shredders in benthic samples (see Section 2.2.5) as a surrogate. Shredders represent a macroinvertebrate functional feeding group that consumes in-stream coarse particulate organic matter (CPOM) [32] and have been used as a proxy of CPOM content [24]. Nutrient (total nitrogen [TN] and total phosphorus [TP]) and bacteria (fecal coliform) sampling occurred once per site during June through August 2018, and May 2019. Grab samples were collected at the downstream end of each reach. All water samples were kept on ice during transport and refrigerated at 4 °C until analyzed. Nutrients were analyzed at NC State University's (NCSU) Center for Applied Aquatic Ecology (CAAE) Lab using the Standard Method 4500 NO3 F EPA Method 353.2 for TN and the Standard Method 4500 P F EPA Method 365.1 for TP within 28 days of sample collection. Fecal coliform grab samples were collected following NC DEQ standards [52] and delivered to NCSU's CAAE Lab within six hours of sample collection for analysis following Standard Method 9222D (MF). Study limitations prevented the collection of consecutive samples over a 5-day period within a 30-day window during the growing season [32]. Thus, one-time samples were collected during the growing season at each site. Instantaneous measurements for summer temperature and specific conductivity were recorded using a YSI Professional Plus Multiparameter Instrument [53] following NC DEQ standards [52] during June through August 2018, and May 2019. In-stream water temperature sensors were not used to record the daily maximum summer temperature at each site, as recommended by Harman and Jones [32], due to project limitations. Rather, one-time temperature measurements were recorded during the summer months (June through August).

### 2.2.5. Biology

Aquatic macroinvertebrate sampling occurred during a two-week period at the end of April and start of May 2018, and a two-day period in April 2019 when taxa of most stream insects are still present in the larval stage. However, two samples were also collected in June and July 2018 due to issues with site access. Samples were collected in accordance with the NC DEQ procedures [54] using the Qual 4 (DAs < 7.8 km$^2$) or Full Scale method (DAs > 7.8 km$^2$), as appropriate. The Full Scale method generates a composite sample from two riffle-kicks, three sweeps, one leaf-pack, two rock- and log-washes, one sand, and one visual assessment, and the Qual 4 method generates a composite sample from one riffle-kick, one sweep, one leaf-pack, and one visual assessment. All specimens were identified to the lowest taxonomic level possible (i.e., genus or species) by an experienced macroinvertebrate biologist. Sampling results were used to calculate four metrics: the EPT richness index, the NC Biotic Index (NCBI), total taxa richness, and % shredders (see Section 2.2.4). EPT focuses on taxonomic orders most sensitive to organic pollutants and stream perturbations [55–57]. EPT richness represents the number of EPT taxa in each sample (i.e., stream site) and has been used to evaluate the success of restoration actions [58]. The NCBI is modeled after the Hilsenhoff Biotic Index [59] and is a weighted average of taxa tolerance values with respect to their abundance [60]. NCBI tolerance values represent a taxon's ability to inhabit stream systems of varying water quality on a scale of 0 (highly intolerant taxa) to 10 (highly tolerant taxa, e.g., [60–62]. The NCBI examines the condition of pollution (level of severity), irrespective of source [60]. A low NCBI indicates low levels of pollution. Total taxa richness is the number of taxa in each sample (i.e., stream site). Unlike total taxa richness, the NCBI and EPT richness indices signal the quality of community composition. Except for two sites (June and July 2018 sites), seasonal correction was applied to the NCBI and EPT richness values to adjust for seasonal patterns of development evident amongst EPT taxa, as samples were collected outside of the summer sampling season (October 1 through May 31) [54]. Benthic organisms were preserved in 95% ethanol for later identification under a stereomicroscope.

### 2.3. Statistical Analyses

### 2.3.1. Response Variable Selection

EPT richness and the NCBI were chosen as response variables because they represent biological processes [63], which are depicted at the top of the Pyramid Framework. According to the Pyramid Framework logic, biological function variable variance should be explained by lower-level variables. Further, these indices are proven to be closely related to changes in ecosystem functions such as leaf litter processing rates, organic matter storage, fine particulate organic matter production and distribution, and secondary production [63].

### 2.3.2. Statistical Approach Selection

To directly test the relative importance of lower-level variables to biological function variables, three linear regression models, including stepwise, lasso, and ridge were applied to the SQT dataset (21 predictor variables, n = 34; Table 3 and reduced versions of the SQT+ dataset (Table 4) to predict EPT richness and the NCBI. Multiple regression approaches were employed because the number of predictors associated with a response a priori is unknown [64]. Engaging multiple modelling approaches and cross-validation to compare model performances can advise which approach is best for a given dataset. Further, prediction of biological data is challenged by implicit uncertainty associated with biological processes, and no single approach has been acknowledged as most valid for such data. Stepwise was chosen because it is a classical method that removes unnecessary predictors that may be redundant and/or add noise to the estimation of other model factors. Lasso (Least Absolute Shrinkage and Selection Operator) and ridge are shrinkage methods that reduce coefficient estimates towards zero to improve model fit [64] while retaining the interpretation of partial regression coefficients [65]. Like stepwise, lasso performs model selection to yield sparse models [64]. Contrary to stepwise, lasso

and ridge can handle highly collinear datasets [66]. Lasso and ridge also perform better in diverging settings. For example, lasso may perform better when the response is a function of few, substantial predictors and ridge may perform better when the response is a function of many predictors, all with partial regression coefficients of similar magnitude [64]. All three models allow for easy interpretation of partial regression coefficients, although the interpretation of ridge models is challenged in settings with many predictor variables [64].

**Table 3.** Partial regression coefficients and coefficients of determination ($R^2$) resulting from stepwise, lasso, and ridge with the SQT dataset for EPT richness and the NCBI. Variables with higher weights are in **bold** (≥|0.20|). Asterisks correspond to values that are statistically significant at applicable alpha levels for stepwise (see below footer).

| Pyramid Level | EPT Richness | | | NCBI | | |
|---|---|---|---|---|---|---|
| Metric | Stepwise[1] | Lasso | Ridge | Stepwise[1] | Lasso | Ridge |
| Hydrology | | | | | | |
|   CN | - | - | −0.03 | - | - | 0.05 |
|   Conc. flow points | - | - | 0.02 | - | - | −0.01 |
|   Soil compaction | - | - | −0.01 | - | - | −0.02 |
| Hydraulics | | | | | | |
|   BHR | - | - | −0.03 | - | - | 0.04 |
|   ER | - | - | 0.02 | - | - | −0.02 |
| Geomorphology | | | | | | |
|   NBS | **−0.48 ***** | −0.19 | −0.06 | **0.24 *** | - | 0.05 |
|   BEHI | **0.23 *** | - | 0.01 | - | - | −0.03 |
|   % Streambank erosion | **−0.33 **** | - | −0.01 | - | - | 0.01 |
|   LWDI | - | - | −0.01 | - | - | 0.00 |
|   Buffer width | - | - | 0.03 | - | - | −0.02 |
|   Basal area | - | - | −0.02 | - | - | 0.04 |
|   Canopy cover | - | - | 0.00 | - | - | 0.00 |
|   Pool depth ratio | **−0.49 ***** | −0.02 | −0.05 | **0.52 ***** | - | 0.07 |
|   % Riffle | - | - | 0.03 | - | - | −0.05 |
|   P-P spacing | - | - | 0.01 | - | - | −0.01 |
|   Sinuosity | - | - | −0.03 | - | - | 0.00 |
| Physicochemistry | | | | | | |
|   TN | - | - | −0.02 | - | - | 0.02 |
|   TP | - | - | −0.02 | **0.20** | - | 0.04 |
|   Fecal coliform | - | - | 0.01 | - | - | 0.01 |
|   Summer temp. | **−0.38 ***** | −0.03 | −0.05 | **0.44 ***** | - | 0.06 |
|   % Shredders | - | - | 0.03 | - | - | −0.05 |
| $R^2$ | 0.64 | 0.37 | 0.64 | 0.53 | - | 0.63 |

[1] "*" α = 0.1; "**" α = 0.05; "***" α = 0.01.

**Table 4.** Partial regression coefficients resulting from stepwise, lasso, and ridge with reduced SQT+ datasets for EPT richness and the NCBI. Variables with higher weights are in **bold** ($\geq$|0.20|). Stepwise was run on the reduced dataset (22 predictors, n = 34). Lasso and ridge were run on the reduced dataset (27 predictors, n = 34). Variables removed from all six models and not listed in the table include: % developed, forested, and impervious cover; bankfull width; bankfull area; BHR; and specific conductivity. N/A indicates that the variable was removed from the model to address multicollinearity and/or dimension reduction. *Italicized* variables are additional variables, not included in the SQT protocol. Asterisks correspond to values that are statistically significant at applicable alpha levels for stepwise (see below footer).

| Pyramid Level | EPT Richness | | | NCBI | | |
|---|---|---|---|---|---|---|
| Metric | Stepwise [1] | Lasso | Ridge | Stepwise [1] | Lasso | Ridge |
| Hydrology | | | | | | |
| *% Agricultural cover* | - | - | 0.02 | **−0.25** | - | −0.04 |
| *DA* | N/A | - | 0.03 | N/A | - | −0.01 |
| CN | - | - | −0.03 | **0.31 \*** | - | 0.04 |
| Conc. flow points | - | - | 0.02 | −0.09 | - | −0.01 |
| Soil compaction | - | - | −0.01 | −0.05 | - | −0.02 |
| Hydraulics | | | | | | |
| ER | **0.37 \*\*\*** | - | 0.02 | - | - | −0.02 |
| *W/D* | **0.40 \*\*\*** | - | 0.03 | - | - | −0.02 |
| $S_{avg}$ | N/A | - | 0.04 | N/A | −0.03 | −0.06 |
| $d_{bkf}$ | **0.27 \*\*\*** | - | 0.02 | 0.06 | - | 0.01 |
| Geomorphology | | | | | | |
| NBS | N/A | −0.15 | −0.05 | N/A | - | 0.04 |
| BEHI | **0.24 \*\*\*** | - | 0.01 | - | - | −0.03 |
| % Streambank erosion | N/A | - | −0.01 | N/A | - | 0.00 |
| LWDI | - | - | −0.01 | - | - | 0.00 |
| Buffer width | **0.28 \*\*\*** | - | 0.03 | - | - | −0.02 |
| Basal area | - | - | −0.02 | - | - | 0.04 |
| Canopy cover | - | - | 0.00 | - | - | 0.00 |
| Pool depth ratio | **−0.25 \*\*\*** | - | −0.04 | - | - | 0.06 |
| % Riffle | **0.49 \*\*\*** | - | 0.03 | - | - | −0.04 |
| P-P spacing | - | - | 0.01 | - | - | −0.01 |
| Sinuosity | - | - | −0.03 | - | - | 0.00 |
| $D_{50}$ | - | 0.03 | 0.04 | - | - | −0.04 |
| $D_{84}$ | **0.41 \*\*\*** | - | 0.03 | - | - | −0.02 |
| Physicochemistry | | | | | | |
| TN | N/A | - | −0.02 | N/A | - | 0.02 |
| TP | - | - | −0.02 | - | - | 0.03 |
| Fecal coliform | - | - | 0.01 | - | - | 0.01 |
| Summer temp. | **−0.59 \*\*\*** | −0.02 | −0.05 | - | - | 0.05 |
| % Shredders | - | - | 0.03 | - | - | −0.05 |
| $R^2$ | 0.88 | 0.38 | 0.75 | 0.18 | 0.21 | 0.70 |

[1] "\*" $\alpha$ = 0.1, "\*\*\*" $\alpha$ = 0.01.

### 2.3.3. Model Implementation

Statistical analyses were performed in RStudio Version 1.2.1335-1 [67]. All predictor and response variables were centered and scaled to account for differences in units between metrics and to allow for comparison between partial regression coefficient magnitudes across response variables. Predictor variables were plotted against response variables to identify outliers.

Variance inflation factors (VIFs) were calculated for predictors to estimate the impact of multicollinearity for the SQT and SQT+ datasets. If multicollinearity was confirmed, data reduction was achieved by iteratively removing predictor variables with the highest VIFs until a VIF threshold of <10 was reached, e.g., [68–71]. Applying the same process, dimension reduction was performed on the highly dimensional SQT+ dataset (34 predictor variables, n = 34) to allow for accurate regression coefficient interpretation [64].

Stepwise was implemented using the regsubsets function in the leaps package in R [72]. Ridge and lasso were implemented using the glmnet package [73]. Residual plots were reviewed for all models to ensure no residual homoscedasticity, linearity, and normality distribution assumptions were violated.

### 2.3.4. Model Selection

Five-fold (5-fold) cross-validation was implemented to select the best models for each approach. For stepwise, the model with the smallest test mean square error was selected (MSE; [64]). If the estimated test error curves were relatively flat, the smallest model for which the estimated test error was within one standard error of the lowest point on the curve was selected [64]. This rule selects the simplest of the relatively equivalent models. The standard error was calculated as follows:

$$\text{SE} = \frac{\sigma}{\sqrt{n}}, \tag{1}$$

where SE equals standard error of the estimated test error across the testing folds for each model size, $\sigma$ equals the standard deviation of the estimated test error across the testing folds for each model size, and $n$ equals the number of folds (five). Cross-validation was implemented using the regsubsets function in the leaps package in R [72].

For lasso and ridge, 5-fold cross-validation was used to find the tuning parameter (t). The tuning parameter controls the strength of the penalty term responsible for shrinking the data. Like stepwise, the one-standard-error rule was applied to find the tuning parameter yielding the most regularized model whose error was within one standard error of the minimum error. Cross-validation was implemented using the cv.glmnet function within the glmnet package in R [73].

### 2.3.5. Model Performance

Test MSEs, generated from 5-fold cross-validation by averaging the MSE across the testing folds, were compared across models to assess predictive performance [64]. The predictive performance was graphically evaluated by comparing the range of predicted values for each model with the range of measured values depicted by box and whisker plots. $R^2$ values were generated for all SQT and reduced SQT+ models by plugging the scaled and centered predictor variable observations into the regression equations to generate predicted values for the SQT and SQT+ full datasets (n = 34). Then, predicted values were plotted against the measured values. $R^2$ values were calculated as follows:

$$R^2 = 1 - \frac{\sum_{i=1}^{n} (y_i - \hat{y}_i)^2}{\sum_{i=1}^{n} (y_i - \overline{y})^2}, \tag{2}$$

where $R^2$ equals the coefficient of determination, $y_i$ equals the predicted value for some value between $y_1$ and $y_{34}$ (n = 34), $\hat{y}_i$ equals the observed value for some value between $y_1$ and $y_{34}$ (n = 34), and $\overline{y}$ equals the mean of the observed data.

## 3. Results

### 3.1. Stream Functional Data for NC Piedmont Ecoregion

The first objective of this paper was to document a range of stream functional data in the Piedmont ecoregion of NC for varied site conditions. Data collected for 37 variables are shown in Table 2. The total number of benthic macroinvertebrate species collected ranged from 6 to 54 species. EPT richness ranged from 0 to 32 species. The NCBI ranged from 2.03 to 8.43.

Total Nitrogen (TN) revealed one extreme outlier and was replaced with the mean value across all sites (n = 34).

### 3.2. Model Implementation

#### 3.2.1. SQT Models

Multicollinearity was moderate (VIFs < 5.0; mean VIF = 3.5) for the SQT dataset; no action was warranted [68–71].

#### 3.2.2. SQT+ Models

Multicollinearity was severe for the SQT+ dataset. Reducing the dataset from 34 to 22 variables (VIFs < 3.9; mean VIF = 2.8; Table 3) was necessary to remove enough linear dependencies to apply stepwise regression. SQT variables and additional variables removed included: % developed cover, % forested cover, % impervious cover, drainage area (DA), bankfull width ($W_{bkf}$), bankfull area ($A_{bkf}$), average channel slope ($S_{avg}$), BHR, % streambank erosion, NBS, TN, and specific conductivity.

Ridge and lasso were performed on the reduced SQT+ lasso/ridge dataset, which was reduced from 34 to 27 variables (VIFs < 8.4; mean VIF = 5.3; Table 4). The variables that were removed included seven SQT and additional variables: % developed cover, % forested cover, % impervious cover, bankfull width ($W_{bkf}$), bankfull area ($A_{bkf}$), BHR, and specific conductivity.

### 3.3. Model Selection

Cross-validation was implemented to select the model for each statistical approach that minimized the mean MSE of the testing folds (Table 5).

**Table 5.** Number of predictor variables and test mean square error (MSE) results following cross-validation of stepwise, lasso, and ridge models used to predict EPT richness and the NCBI from two data sets (SQT and SQT+).

| | | EPT Richness | | | NCBI | | |
|---|---|---|---|---|---|---|---|
| | | Stepwise | Lasso | Ridge | Stepwise | Lasso | Ridge |
| SQT | # of predictor variables | 5 | 3 | 21 | 4 | 0 | 21 |
| | Test MSE | 1.24 | 1.40 | 1.26 | 1.46 | 1.26 | 1.26 |
| SQT+ | # of predictor variables | 9 | 3 | 22 | 5 | 1 | 27 |
| | Test MSE | 0.79 | 1.33 | 1.18 | 1.17 | 1.26 | 1.01 |

### 3.4. Model Performance

### 3.4.1. SQT Models

The range in measured EPT richness was best predicted by the stepwise model (Figure 3a), which moderately captured the range of measured values and matched the measured mean. Lasso and ridge models failed to capture the range of measured EPT richness, and the mean values were slightly higher than the measured mean. Stepwise and ridge both resulted in moderate predictions of EPT richness ($R^2$ = 0.64; Table 3). However, stepwise was the best predictive, parsimonious model.

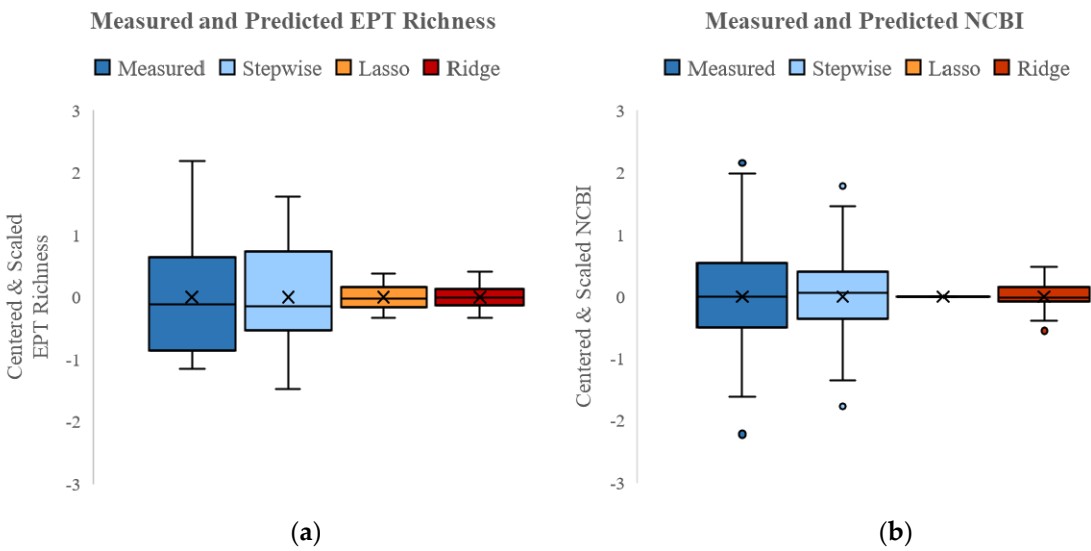

**Figure 3.** Measured and predicted centered and scaled values for (**a**) EPT richness and (**b**) NCBI for the SQT dataset (n = 34). The "x" indicates the mean value.

Similarly, the range in the measured NCBI was best predicted by the stepwise model (Figure 3b). The stepwise model exhibited a similar mean value as the measured mean but a slightly smaller range of predicted values. Lasso and ridge models failed to capture the range of the measured NCBI. Lasso did not include any variables in the model, meaning that the intercept alone best predicted the NCBI. Ridge resulted in the best prediction of the NCBI ($R^2$ = 0.63; Table 3) followed by stepwise ($R^2$ = 0.53; Table 3). However, stepwise was the best predictive, parsimonious model.

To understand which variables are most important to predicting EPT richness and the NCBI, the partial regression coefficients were compared across model approaches (Table 3). Variables with the largest coefficients (positive or negative) have the greatest influence on prediction. All predictors included in the stepwise models were ≥|0.20|. Substantial negative predictors (≥−0.20) with the greatest influence on EPT richness prediction include NBS, % streambank erosion, pool depth ratio, and summer temperature. BEHI was the only positive predictor (≥0.20) with the greatest influence on EPT richness. Substantial positive predictors (≥0.20) for the NCBI included NBS, pool depth ratio, TP, and summer temperature. NBS, pool depth ratio, and summer temperature were important in predicting both the NCBI and EPT richness.

For both ridge models, the magnitudes of all partial regression coefficients were less than |0.07|. For both ridge models, canopy cover had a marginal effect on EPT richness and the NCBI, and LWDI and sinuosity had marginal effects on the prediction of the NCBI.

### 3.4.2. SQT+ Models

The reduced SQT+ stepwise model predicted EPT richness better than lasso and ridge. The stepwise mean was slightly lower than the measured mean, and the range of predicted values was similar to the range of measured values (Figure 4a). Again, the lasso and ridge models had similar means to the measured mean, but the ranges were substantially smaller than the range of measured values. Stepwise resulted in the highest prediction of EPT richness ($R^2$ = 0.88; Table 4), followed by ridge ($R^2$ = 0.75; Table 4).

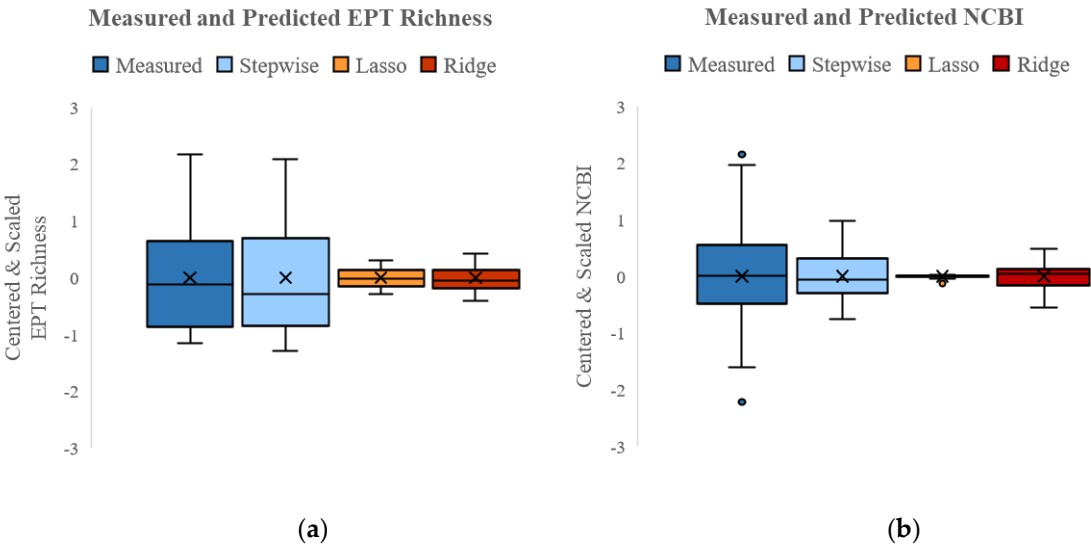

**Figure 4.** Measured and predicted centered and scaled values for (**a**) EPT richness and (**b**) the NCBI for the reduced SQT+ stepwise dataset (n = 34). The "x" indicates the mean value.

All three NCBI models reported ranges that were smaller than the measured range (Figure 4b). Ridge greatly outperformed stepwise and lasso in prediction ($R^2$ = 0.70; Table 4). Stepwise had the largest range of predicted values, which more closely matched the range of measured values, but the lowest $R^2$ of all three models.

All predictors included in the stepwise EPT richness models were ≥|0.20|, and the variables with the greatest negative influence were pool depth ratio and summer temperature. Substantial positive predictors were ER, W/D, $d_{bkf}$, buffer width, % riffle, and $D_{84}$. For the stepwise NCBI model, % agricultural cover had the greatest negative influence and CN had the greatest positive influence on the NCBI.

For both ridge models, the magnitudes of all partial regression coefficients were relatively small (≤|0.06|). For both ridge models, canopy cover had zero influence (i.e., partial regression coefficient equals zero) on EPT richness and the NCBI, and % streambank erosion, LWDI, and sinuosity had zero influence on prediction of the NCBI.

## 4. Discussion

### 4.1. SQT Protocol Biotic Prediction and Pyramid Framework Premise

Stepwise and ridge both yielded robust predictions for EPT richness and the NCBI for the SQT dataset, demonstrating that the SQT protocol can reasonably predict biological function. The ridge models provided direct evidence that the SQT protocol supports the Pyramid Framework: variables from all levels of the Pyramid were important to biological function, where each single variable was relatively equal in importance to predicting biology (partial regression coefficients ranged from |0.00 to 0.07|). In contrast, the stepwise models included variables from the geomorphic and physicochemical Pyramid levels, suggesting that reach-scale variables were more important to biological function than watershed hydrology variables. However, the functions depicted in the Pyramid Framework are organized hierarchically by interrelatedness. Hydrological variables are depicted at the bottom of the Pyramid because they directly affect the greatest number of other function variables. Therefore, it is most likely that hydrologic variables directly or indirectly influenced higher-level variables (i.e., hydraulic, geomorphology, and physicochemical), and that reach-scale hydraulic, geomorphology, and physiochemical variables may more directly represent the effect of watershed condition on the streams. It is also possible that hydrology failed to explain a large variance in biological function because the SQT protocol lacks hydrological variables pertinent to macroinvertebrates.

Interestingly, the SQT dataset did not require variable reduction, suggesting that metrics included in the SQT protocol each provide unique information. Meanwhile, the SQT+ dataset exhibited high multicollinearity, which required the removal of twelve variables for stepwise, and the removal of seven variables for lasso and ridge. The elimination of channel incision (BHR), bank erosion stress potential (NBS), active streambank erosion (% streambank erosion), and local water quality (TN) indicates redundancy in the SQT protocol in the presence of the following additional metrics: % agricultural cover, drainage area (DA), average channel slope ($S_{avg}$), mean bankfull depth ($d_{bkf}$), channel shape (W/D), and substrate size ($D_{50}$ and $D_{84}$). Overall, the inclusion of these additional metrics improved EPT richness prediction for stepwise and ridge, and the NCBI for ridge, suggesting that the SQT protocol is failing to document some factors important to macroinvertebrates. Thus, to improve the biotic prediction of the SQT protocol, developers should consider including these additional variables either as SQT metrics or as stratifying variables for reference curves, which relate measured field and desktop values to function index scores in the SQT excel workbook tool [74].

### 4.2. Variables Important to Macroinvertebrates

Detailed discussion of model variables was limited to the best performing and/or parsimonious models across both datasets (i.e., stepwise SQT and SQT+ models for EPT richness; and stepwise SQT and ridge SQT+ models for the NCBI).

Stepwise regression with the SQT dataset indicated that bank erosion related variables (BEHI, NBS, and % streambank erosion), pool depth ratio, and local water quality (summer temperature) explained the variability in EPT richness. Stepwise regression with the SQT+ dataset predicted EPT richness with nine variables: floodplain width (ER), channel shape (W/D), mean bankfull depth ($d_{bkf}$), bank erosion susceptibility (BEHI), riparian buffer width, pool depth ratio, extent of riffle and run habitat (% riffle), substrate size ($D_{84}$), and summer temperature. The stepwise SQT+ model included all SQT variables that were also significant predictors ($\alpha = 0.1$) for the SQT model, except for NBS and % streambank erosion, which were not retained in the reduced SQT+ dataset due to high multicollinearity. The positive relationships between mean bankfull depth ($d_{bkf}$), floodplain width (ER), and substrate size ($D_{84}$) to EPT richness reinforces findings by Doll et al. [24], which suggested that channel size, floodplain width, and substrate should be a focus for restoration projects that aim to improve macroinvertebrate community composition and biological condition. The positive relationship between % riffle and EPT richness confirms that riffles, which exhibit heterogeneity, stability, and loosely packed substrate which facilitate higher oxygen concentrations, are critical habitat for EPT

taxa [13]. Further, the negative relationship between pool depth ratio and EPT richness exhibited in both the SQT and SQT+ models confirms EPT taxa habitat excludes pools, as they are typically sampled from riffles, leaf packs, sticks, logs, roots, and other areas of debris, especially along the streambanks [54]. Temperature was negatively related to EPT richness for both the SQT and SQT+ models, confirming that elevated stream temperatures negatively affect sensitive taxa and supporting findings by Wang and Kanehl [75] for 39 Minnesota and Wisconsin streams.

Wide and shallow channels (indicated by high W/D) were associated with high EPT richness (Table 4), which supports findings presented by Nerbonne and Vondracek [76] but contradicts findings by Doll et al. [24]. Generally, over-wide and shallow streams are considered less supportive of biota such as fish and macroinvertebrates due to decreased fish habitat (shallow pools) and decreased macroinvertebrate food supply. W/D is calculated by dividing bankfull width ($W_{bkf}$) by mean bankfull depth ($d_{bkf}$), and thus will generally increase with stream size. Because drainage area (DA) was removed from the SQT+ dataset for stepwise to reduce the effect of multicollinearity, W/D may be serving as a surrogate for stream size (DA). Stream size has been shown to be positively related to higher numbers of total taxa and EPT taxa [77] and a driver for macroinvertebrate communities [25]. Further, mean bankfull depth ($d_{bkf}$) was also positively related with EPT richness (Table 4). This finding concurs with findings from Doll et al. [24] and conflicts with the association of high EPT richness with wide and shallow channel shapes in this study, providing additional evidence that W/D is likely representing stream size in EPT richness SQT+ models.

Surprisingly, high bank erosion susceptibility (BEHI) was associated with high EPT richness in the SQT and SQT+ models, suggesting that an increase in EPT taxa is positively associated with an increase in bank erosion. Because BEHI scores ranged between "low" and "high," with most sites scoring as "moderate" (n = 21), it is possible that the results demonstrate the benefits of natural erosional processes, which create habitat and offer refugia for biota during high flows [78,79]. However, interpreting the influence of BEHI is difficult because it is an index comprised of five variables that are individually rated at each eroding bank and meandering bend, of which the dominant score is used to characterize erosion potential for the reach [32]. Furthermore, BEHI evaluates the potential for erosion, not active erosion. For example, a streambank that is not actively eroding could be rated as "moderate" erosion potential if it exhibited little vegetative cover, low root density, and a steep bank angle, all of which could be characteristic of alluvial meandering streams located in highly forested riparian corridors. Additionally, the positive relationship between BEHI and EPT richness could be related to the lack of variability in dominant BEHI scores. However, the positive relationship between BEHI and EPT richness is also contradicted by the negative relationships between streambank erosion and NBS with EPT richness in the SQT model, which indicate that channels with less bank erosion (extent and magnitude) are more favorable for macroinvertebrates. Further, the relationship challenges Simpson et al. [80], who found that banks with low BEHI were associated with more taxonomically rich and spatially stable macroinvertebrate assemblages, and banks with higher BEHI were associated with less taxonomically rich and spatially variable macroinvertebrate assemblages. Taken together, the relationship between streambank erosion (BEHI, NBS, % streambank erosion) and macroinvertebrates (EPT taxa) is complex. The resolution offered by one dominant BEHI score per stream site is likely insufficient to characterize the ideal bank erosion preferred by macroinvertebrate communities.

Due to the variety of restored and degraded sites set in urban and agricultural watersheds (Table 1), we anticipated strong negative relationships between land use cover and macroinvertebrate metrics. High multicollinearity precluded most additional hydrologic variables (% impervious, forested, and developed cover) from the SQT+ models. However, the prevalence of multicollinearity reiterates the same occurrence exhibited by the SQT models: hydrologic variables may be directly or indirectly influencing higher-level variables, which may more directly represent the effect of watershed condition on the streams. This is not surprising, given that land use change integrates many anthropogenic activities that cumulatively, negatively affect stream and riparian ecosystems, serving as a general index for human-induced disturbance [13]. Furthermore, specific conductivity, an important water

quality predictor for assessing macroinvertebrate community integrity [81–84], was also removed from the model due to significantly ($\alpha$ = 0.05) collinear relationships exhibited with % forested cover, CN, summer temperature, and mean bankfull depth (Figure A1). This implies that specific conductivity was represented by other variables in the model.

Interestingly, the stepwise prediction of the NCBI was better for the SQT dataset compared to the SQT+ dataset, but ridge predicted the NCBI better for the SQT+ dataset. The exclusion of NBS from the reduced stepwise SQT+ dataset, a substantial predictor of the NCBI for the SQT dataset, may explain why the prediction of the NCBI was reduced. Stepwise regression with the SQT dataset indicated that bank erosion stress potential (NBS), pool depth ratio, and local water quality (TP and summer temperature) positively influenced the NCBI. Higher NCBI scores correspond to degraded, pollutant-tolerant macroinvertebrate communities. Therefore, as NCBI values increase, biotic condition declines. Surprisingly, pool depth ratio had the greatest influence, with the NCBI generally increasing as pool depth ratio increased. High bank erosion stress (NBS) was also positively related to the NCBI, indicating that streams with high bank stress potential exhibit a negative effect on biological function.

The ridge model, containing 27 variables, best explained variability amongst the NCBI for the SQT+ dataset. Variables with the greatest partial regression coefficients ($\geq$|0.05|) include average channel slope ($S_{avg}$), pool depth ratio, summer temperature, and organic carbon (% shredders). Pool depth ratio and summer temperature were also substantial predictors for the NCBI for the SQT dataset.

## 5. Conclusions

To gain insight into the applicability of the conceptual Pyramid Framework as the premise of the SQT, the SQT protocol was implemented in Piedmont, NC (Southeast, USA). Statistical analyses revealed stepwise and ridge as the best predictive models for the biotic integrity data collected in this study. The best performing stepwise and ridge models identified key variables that influence stream macroinvertebrates communities: floodplain width (ER), channel shape (W/D), channel dimension ($d_{bkf}$), bank erosion stress and susceptibility (NBS and BEHI), active streambank erosion, pool depth ratio, buffer width, adequate extent of riffle and run habitat (% riffle), substrate size ($D_{84}$), and stream summer temperature. Results suggest that restoration activities such as increasing floodplain and buffer widths, planting riparian vegetation to shade streams, and reducing streambank erosion and erosion potential will improve macroinvertebrate community composition. Results also suggest that creating appropriate local conditions such as an ample riffle habitat with a suitable substrate type will help recruit and support EPT taxa, although the presence of contaminants [60] and the existence and diversity of the regional species pool [13] are also critical factors for improving community composition. Nevertheless, practitioners should take care to design streams that will support not only adequate riffles, but also other habitat units such as pools, stable undercut banks, leaf packs, overhanging vegetation, and point bars, to support a diversity of aquatic species. Our findings help elucidate the complex relationships between watershed- and reach-scale variables and biological function and can be used to help practitioners establish feasible restoration goals with appropriate success metrics to improve stream restoration success.

Statistical analyses revealed that the SQT protocol reasonably predicts biological function, and cross-validation and comparisons of model performance indicated that stepwise and ridge models should reasonably predict EPT richness and the NCBI in NC and the greater southeast Piedmont (USA). Results provided moderate support for the hierarchical Pyramid Framework: highly predictive ridge models included metrics from all Pyramid levels, while highly predictive stepwise models included metrics from higher Pyramid levels, excluding watershed hydrology variables. Reach-scale metrics were more important than watershed hydrology metrics to predict macroinvertebrates, suggesting that successful biological functional restoration is possible despite watershed condition. However, the removal of several additional watershed variables from the SQT+ models, due to multicollinearity, underscores that watershed variables directly and indirectly affect higher-level variables, reinforcing relationships depicted in the Pyramid Framework. Further, that so few variables were needed to

predict biological function suggests that importance should be placed on measuring these metrics in the NC and southeast Piedmont. Considering limited time and money, practitioners can focus efforts on measuring the most important metrics to understand the integrity of the biotic community and to inform restoration goals and design decisions.

Although prediction of macroinvertebrate biota with the SQT protocol was reasonable, the fact that 36% to 47% of variability amongst EPT richness and the NCBI remains unexplained by the best SQT models and that the SQT+ models improved predictions for most models suggests the SQT protocol is lacking critical measures of function. Future endeavors to repeat this approach with (1) a larger sample size with equal representation of restored, degraded, and geomorphic and biological reference sites; (2) robust monitoring data, especially for highly variable physiochemical parameters; (3) additional physicochemical variables, such as biochemical oxygen demand and pH; and (4) fish and other fauna as indicators of biological function, will augment findings.

The SQT excel workbook tool applies arbitrary, implicit weighting by averaging all metrics within each function category (e.g., hydrology). As the number of metrics within a function category (e.g., geomorphology) increases, the significance that each metric carries decreases. Improved regression models could result in multipliers for individual metrics that more accurately reflect their importance to the biological function, thus improving the accuracy of the overall function quantification. However, the SQT allows users some choice in metric and assessment method selection. This flexibility challenges the creation of multipliers because the relationship of each metric to biological function will change with respect to other metrics measured. Thus, as the number of metrics change, the ability of the SQT protocol to reasonably predict biological function will also change.

Regional testing of the SQT in other states resulted in revisions, additions, and deletions of some metrics and assessment methods [33–36]. Thus, this paper lays out an approach to evaluate the Pyramid Framework and test regionalized versions of the SQT protocol. Further, our results can help inform updates to the NC SQT protocol [32], which is currently under revision.

Our findings indicate that the SQT displays promise and merit. With further refinement, the SQT can be a beneficial tool for practitioners and regulators. The future of stream restoration remains hopeful, as conceptual frameworks and tools for practitioners and regulators, like the SQT, are created and evaluated to ultimately improve the practice of stream restoration.

**Author Contributions:** Conceptualization, B.D. and J.P.; data curation, S.D.; formal analysis, S.D.; funding acquisition, B.D. and J.P.; investigation, S.D., B.D., and J.P.; methodology, S.D., B.D., and J.P.; project administration, B.D.; supervision, B.D.; validation, S.D. and N.N.; writing—original draft, S.D.; writing—review and editing, B.D. and N.N. All authors have read and agreed to the published version of the manuscript.

**Funding:** This research was funded in part by the NC Department of Environmental Quality Division of Mitigation Services and the Environmental Defense Fund. Funding has not been provided to cover the costs to publish in open access.

**Acknowledgments:** Many thanks to: Zahra Saki, Megan Moore, Emily Griffith, and Jason Osborne for statistical consulting; Dave Penrose and Jason York for macroinvertebrate data collection, identification, and thoughtful discussion; Cameron Jernigan for data collection and data processing assistance; and Jack Kurki-Fox for data processing assistance.

**Conflicts of Interest:** The authors declare no conflict of interest.

## Appendix A

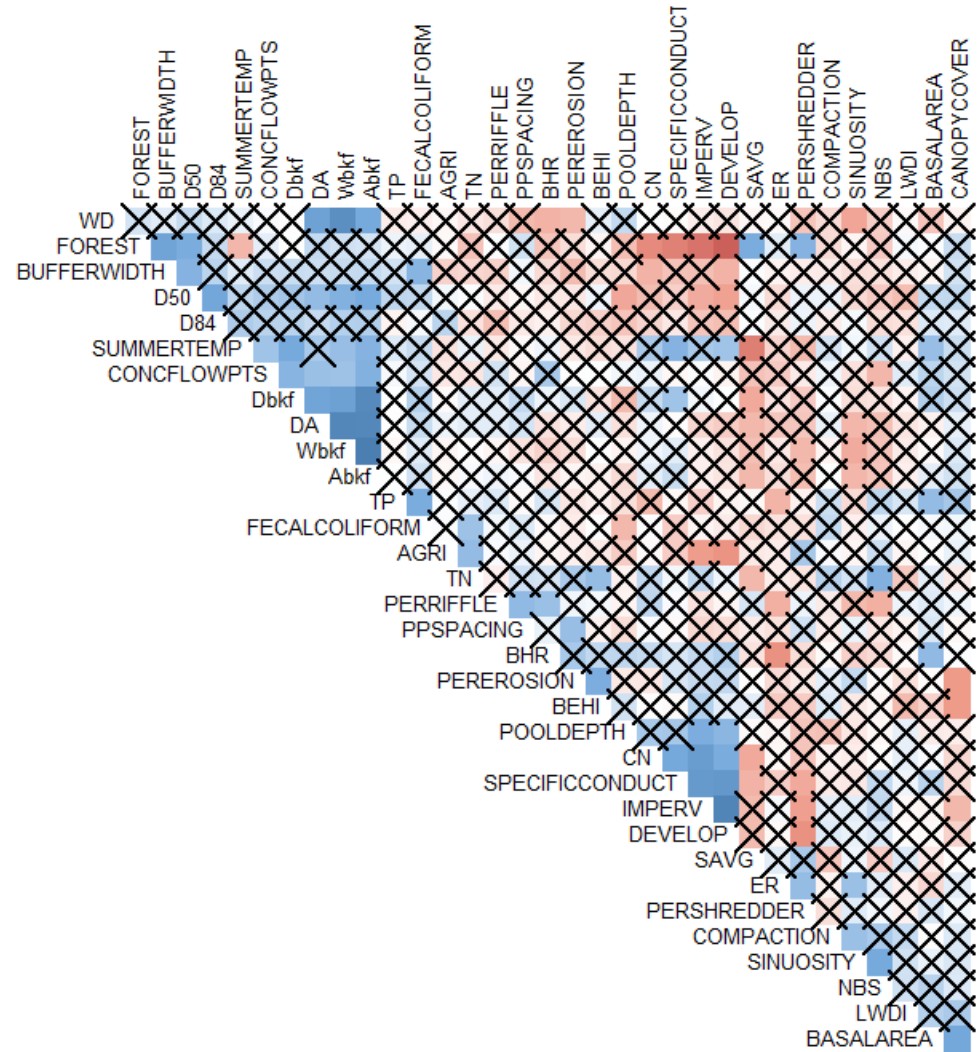

**Figure A1.** Correlation matrix for the SQT+ dataset, where insignificant pairwise correlations at the α = 0.05 level are demarcated with an "x." Red indicates negative pairwise correlations and blue indicates positive pairwise correlations along a color gradient, whereby the correlation coefficient increases as the red and blue colors darken.

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
