# Peer review of "Can the Stream Quantification Tool (SQT) Protocol Predict the Biotic Condition of Streams in the Southeast Piedmont (USA)?"

_water, doi:10.3390/w12051485_

Round 1
Reviewer 1 Report
The paper by Donatich et al. tests the ability of the SQT to predict ecological function. Overall the paper is very well written and my comments address specific parts of the paper that could be improved.
- The authors state several objectives and hypotheses in the introduction. The discussion would be improved if they returned more directly to these objectives and summarized the results in this context. By doing this the discussion would have more structure. Currently the discussion reads as a rewording of the results section.
- There is a lack of information on where the study sites are in the watershed. What is the upstream environment like for these sites? How was the land use distributed across the sites? That is - were the restored sites urban or agricultural or both? Were the degraded sites urban? Would the authors expect there to be different relationships in the models between function and the SQT based on land use? A discussion of this potential variability would enhance the discussion.
- The 2011 NCLD was used for quantifying land use. Has land use in some of these watersheds changed since that time? Often in this region land use changes quickly and what was forest or agriculture is now urban.
- I believe the summer temperature measures are one of the weakest inputs to the model. Most data were collected in April and 2 in June/July. In the NC Piedmont there can be a large temperature difference between these months which can also influence the life history of the organism you collect. Thus, the temperature relationship might be an artifact of the sampling. Can the authors address this potential impact?
- Why were the partial regression coefficients set at +/- 0.20?
- What is the significance that so few variables were needed to predict function?
- Given that hydrology was not important in any model, and hydrology was quantified as CN, can the authors discuss whether a different hydrology variable might more informative for the model? What is the advantage of using CN over other variables?
- I found it interesting that conductivity was not a major predictor variable in the model but TP was. Given that conductivity is one of the best water quality predictors for macroinvertebrates in other studies, can the authors clarify why they think it was not important in their model?
- In line 497-499 the authors make a recommendation for variables that should be focus of restoration for macroinvertebrates. The authors should add a clarifying statement that other variables are important for macroinvertebrate recovery such as the presence of other contaminants or source populations.
- In line 502 the authors state pools are not important habitat for EPT. I agree; however, the paper would be strengthened by expanding the conclusion that pools are important habitat for other invertebrates that bring additional functions (or even redundant) functions to the ecosystem. A stream restored to nothing but riffles does not necessarily have a high degree of functioning.
- In line 568-570; I am not completely convinced your paper allows you to reach this conclusion, it seems a bit of a reach. The model removed a lot of variables and did not explain most of the variability suggesting there is a lot we do not know about predicting function.
- From a testing standpoint the point of the paper is clear. However, how does knowing these relationships advance stream restoration science and practice? The discussion would be improved by taking a larger step out and placing the research in context.
Author Response
FYI: Per our assigned managing editor, Reviewer 2 comments were addressed directly in the revised manuscript. We provided point-by-point responses to Reviewer 1 (your) comments below, and refer to lines in the revised manuscript attached.
Point 1: The authors state several objectives and hypotheses in the introduction. The discussion would be improved if they returned more directly to these objectives and summarized the results in this context. By doing this the discussion would have more structure. Currently the discussion reads as a rewording of the results section.
Response 1: The authors aimed to frame the discussion by creating two headers:
- SQT Protocol Biotic Prediction & Pyramid Framework Premise (Objective 2)
- Variables important to macroinvertebrates (Objective 3)
While addressing all other comments in this 5-day period, it has not been possible to completely rework the discussion. Several of the following points by Reviewer 1 also referred to the discussion and conclusion. In directly addressing these points, the authors believe they have improved the discussion section, and placed results in a larger context in the conclusion section.
Point 2: There is a lack of information on where the study sites are in the watershed. What is the upstream environment like for these sites? How was the land use distributed across the sites? That is - were the restored sites urban or agricultural or both? Were the degraded sites urban? Would the authors expect there to be different relationships in the models between function and the SQT based on land use? A discussion of this potential variability would enhance the discussion.
Response 2: The authors have added a table (Table 1) of data organized by stream condition (e.g. restored) including: watershed size, channel slope, land use (impervious, developed, forested, and agricultural cover), and median particle size to provide more information on the study sites. The study sites are headwater streams (orders range from 1-3).
Point 3: The 2011 NCLD was used for quantifying land use. Has land use in some of these watersheds changed since that time? Often in this region land use changes quickly and what was forest or agriculture is now urban.
Response 3: Your point is relevant for one restored and degraded sites, in which active development was occurring in the watershed. The other 32 watersheds are relatively stable. Unfortunately, this was the best data available at the time of data collection.
Point 4: I believe the summer temperature measures are one of the weakest inputs to the model. Most data were collected in April and 2 in June/July. In the NC Piedmont there can be a large temperature difference between these months which can also influence the life history of the organism you collect. Thus, the temperature relationship might be an artifact of the sampling. Can the authors address this potential impact?
Response 4: Stream temperature data was collected during June through August 2018, and May 2019. The authors reviewed historical weather data from Wunderground by locating the nearest station available to the site and documented the low and high temperatures on days of sampling and the daily average temperature. Below is a summary by month and year of WQ sampling events.
|
Month, Year |
Range (Low to High; deg. C) |
Average of daily average temperatures on sampling days (deg. C) |
|
June, 2018 |
16.7-32.2 |
24.3 |
|
July, 2018 |
12.8-33.9 |
25.2 |
|
August, 2018 |
21.1-32.8 |
25.9 |
|
May, 2019 |
16.7-33.3 |
24.8 |
|
June, 2019 |
20-32.2 |
24.0 |
The data shows that temperatures in these months and across years were very similar, thus we do not think the strong relationship we observed in the models between stream temperature and macroinvertebrates is a residual artifact. Further, macros were collected in April and May, except for two sites (which were applied a seasonal correction to account for “strong seasonal patterns of development, with emergence tied to photoperiod and temperature” (NC DEQ SOP 2016). Any potential artifacts between temperature and macros, we believe was also addressed by applying a seasonal correction to the NCBI and EPT richness indices.
NC Department of Environmental Quality. Standard Operating Procedures for the Collection and Analysis of Benthic Macroinvertebrates. 2016.
Point 5: Why were the partial regression coefficients set at +/- 0.20?
Response 5: The authors used |0.20| as a threshold to aid in results interpretation and discussion. However, if the editor feels strongly that we should remove this threshold, then we can refer to partial regression coefficient as higher or lower and not include a specific threshold.
Point 6: What is the significance that so few variables were needed to predict function?
Response 6: Importance should be placed on measuring these metrics. Considering limited time and money, practitioners can focus efforts on measuring these important variables to understand the integrity of the biotic community and to inform restoration goals and design decisions.
This point was added to the Conclusion: lines 617-619.
Point 7: Given that hydrology was not important in any model, and hydrology was quantified as CN, can the authors discuss whether a different hydrology variable might more informative for the model? What is the advantage of using CN over other variables?
Response 7: CN was included because it is an SQT hydrology metric. The authors considered that other hydrologic metrics might be important to biological function (based on the literature), which is why the SQT+ dataset included % developed, forested, agricultural, and impervious cover. CN takes both soils and land use into account, but the literature provides strong evidence for connection between land use (such as % developed, forested, agricultural, and impervious cover) and macroinvertebrates (see introduction).
Of the additional variables added, only % agriculture was substantial in the SQT+ models. However, % impervious, developed cover, and forested cover had to be removed due to severe multicollinearity.
CN was relatively important predictor (compared to importance of other predictors in the model) for NCBI in the ridge model for the SQT dataset. For the SQT+ dataset, CN and % agricultural cover were relatively important predictors.
Point 8: I found it interesting that conductivity was not a major predictor variable in the model but TP was. Given that conductivity is one of the best water quality predictors for macroinvertebrates in other studies, can the authors clarify why they think it was not important in their model?
Response 8: Due to multicollinearity, specific conductivity was removed from the SQT+ models (it is not included in the SQT model). The removal of this variable does not imply that it is not an important predictor; rather its value is collinear to other variables that were retained in the models for this study. The correlation matrix for the SQT+ dataset reveals that specific conductivity was significantly correlated (at alpha =0.5 level) with % forested cover, summer temperature, mean bankfull depth, and CN, several of which were important to predicting EPT and BI.
This point was added to the Conclusion: lines 564-569.
Point 9: In line 497-499 the authors make a recommendation for variables that should be focus of restoration for macroinvertebrates. The authors should add a clarifying statement that other variables are important for macroinvertebrate recovery such as the presence of other contaminants or source populations.
Response 9: The authors agree with the Reviewer’s statement. The authors refer you to Response 10.
Point 10: In line 502 the authors state pools are not important habitat for EPT. I agree; however, the paper would be strengthened by expanding the conclusion that pools are important habitat for other invertebrates that bring additional functions (or even redundant) functions to the ecosystem. A stream restored to nothing but riffles does not necessarily have a high degree of functioning.
Response 10: The authors agree with the Reviewer’s concern. The authors felt it was more appropriate to address this concern in the conclusion section: lines 596-602.
Point 11: In line 568-570; I am not completely convinced your paper allows you to reach this conclusion, it seems a bit of a reach. The model removed a lot of variables and did not explain most of the variability suggesting there is a lot we do not know about predicting function.
Response 11: For biological systems, the authors believe the R2 values are reasonable. The authors refer to an example study (Arhonditsis & Brett, 2004; Table 1), which demonstrates that in a review of 153 peer-reviewed studies, the median model performance was usually less than 0.65 for the biological variables.
The authors acknowledge in the conclusion section that 36 to 47% of variability unexplained is still substantial, and highlights that the SQ T is therefore missing important metrics – which is evident by the improved prediction exhibited by the SQT+ dataset.
Arhonditsis, G. B., & Brett, M. T. (2004). Evaluation of the current state of mechanistic aquatic biogeochemical modeling. Marine Ecology Progress Series, 271, 13-26.
Point 12: From a testing standpoint the point of the paper is clear. However, how does knowing these relationships advance stream restoration science and practice? The discussion would be improved by taking a larger step out and placing the research in context.
Response 12: The authors agree with the Reviewer’s point. The authors have added additional text to the Conclusion to place results in a larger context. See lines 602-604; 615-619; and 640-641.

Reviewer 2 Report
Dear Dr. Donatich and co-authors,
I have now carefully read your MS entitled: “Can the Stream Quantification Tool (SQT) protocol predict the biotic condition of streams in the Southeast Piedmont (USA)?”. The article objective is to find significant relationships between a number of environmental variables (in a hierarchical multi-scaled framework, from watershed-scale hydrology to reach scale local water quality) and two biological integrity metrics, treated as surrogates of stream ecosystem overall functioning. Authors choose the widely used EPT richness and the regional biotic index NCBI. Using data from 34 headwater streams (small watersheds, up to 22 km2 and 1st to 3rd order streams), different multi-parameter predictive models –using three regression techniques- are constructed, compared and discussed.
I would highlight the interest of this investigation, as testing the value as surrogate of biological condition (function) of environmental assessments such as the SQT (Harman et al. 2012), could add confidence on these kind of approaches and make them promising tools for a first screening of general conservation and restoration needs in stream ecosystems. Thus, the results of this study support the use of a simple protocol based on easily measurable environmental variables to predict the ecological integrity (or biological function) of certain stream reaches of the Piedmont ecorregion in SW US. As the authors acknowledge, further investigations are desirable to expand the geographical content of the study and extrapolate the use of this protocol as a functional assessment of stream condition to other ecorregions.
Overall, the paper is well written, methods are well justified and results are presented with clarity and thoroughly discussed.
I have made some general and specific comments through the MS itself.
A commented PDF file is attached.
I hope my comments and suggestion can be useful to gain interest and to facilitate the readability and understandability of the paper.
Sincerely,
**Harman, W., R. Starr, M. Carter, K. Tweedy, M. Clemmons, K. Suggs, C. Miller. 2012. A Function-Based Framework for Stream Assessment and Restoration Projects. US Environmental Protection Agency, Office of Wetlands, Oceans, and Watersheds, Washington, DC EPA 843-K-12-006.

Author Response
Responses to Reviewer 2's comments are directly addressed in the revised manuscript.

Round 2
Reviewer 1 Report
Thank you for addressing comments.
Reviewer 2 Report
Dear Sara and coauthors,
I have now reviewed the new version of your ms, as well as read your responses to my questions/suggestions and I think that the paper has improved in interest and readability. Thanks for that and congrats for your work!